# Mexican Colorectal Cancer Research Consortium (MEX-CCRC): Etiology, Diagnosis/Prognosis, and Innovative Therapies

**DOI:** 10.3390/ijms24032115

**Published:** 2023-01-20

**Authors:** Antonio Andrade-Meza, Luis E. Arias-Romero, Leonel Armas-López, Federico Ávila-Moreno, Yolanda I. Chirino, Norma L. Delgado-Buenrostro, Verónica García-Castillo, Emma B. Gutiérrez-Cirlos, Imelda Juárez-Avelar, Sonia Leon-Cabrera, Mónica G. Mendoza-Rodríguez, Jonadab E. Olguín, Araceli Perez-Lopez, Carlos Pérez-Plasencia, José L. Reyes, Yesennia Sánchez-Pérez, Luis I. Terrazas, Felipe Vaca-Paniagua, Olga Villamar-Cruz, Miriam Rodríguez-Sosa

**Affiliations:** 1Unidad de Investigación en Biomedicina, Facultad de Estudios Superiores-Iztacala (FES-I), Universidad Nacional Autónoma de México (UNAM), Tlalnepantla 54090, Mexico; 2Programa de Doctorado en Ciencias Biomédicas, Universidad Nacional Autónoma de México (UNAM), Ciudad de México 04510, Mexico; 3Programa de Doctorado en Ciencias Biológicas, Universidad Nacional Autónoma de México (UNAM), Ciudad de México 04510, Mexico; 4Carrera de Médico Cirujano, Facultad de Estudios Superiores-Iztacala (FES-I), Universidad Nacional Autónoma de México (UNAM), Tlalnepantla 54090, Mexico; 5Laboratorio Nacional en Salud: Diagnóstico Molecular y Efecto Ambiental en Enfermedades Crónico-Degenerativas, Facultad de Estudios Superiores-Iztacala (FES-I), Universidad Nacional Autónoma de México (UNAM), Tlalnepantla 54090, Mexico; 6Subdirección de Investigación Básica, Instituto Nacional de Cancerología, Ciudad de México 14080, Mexico

**Keywords:** Mexican Colorectal Cancer Research Consortium (MEX-CCRC), colorectal cancer, colorectal cancer diagnosis, colorectal cancer prognosis, colorectal cancer therapy, antitumor immune response

## Abstract

In 2013, recognizing that Colorectal Cancer (CRC) is the second leading cause of death by cancer worldwide and that it was a neglected disease increasing rapidly in Mexico, the community of researchers at the Biomedicine Research Unit of the Facultad de Estudios Superiores Iztacala from the Universidad Nacional Autónoma de México (UNAM) established an intramural consortium that involves a multidisciplinary group of researchers, technicians, and postgraduate students to contribute to the understanding of this pathology in Mexico. This article is about the work developed by the Mexican Colorectal Cancer Research Consortium (MEX-CCRC): how the Consortium was created, its members, and its short- and long-term goals. Moreover, it is a narrative of the accomplishments of this project. Finally, we reflect on possible strategies against CRC in Mexico and contrast all the data presented with another international strategy to prevent and treat CRC. We believe that the Consortium’s characteristics must be maintained to initiate a national strategy, and the reported data could be useful to establish future collaborations with other countries in Latin America and the world.

## 1. The Problem

The CRC ranks third in incidence and is the second leading cause of death by cancer worldwide, with more than 1.9 million cases estimated for 2020 [1]. Most epidemiologists consider that CRC cases will reach 3.2 million by 2040 [2]. 

In Mexico, the CRC mortality rate has increased annually by 2.7% for men and 1.3% for women [3]. Notably, the mortality records associated with CRC for both sexes show that from 1990 to 2020, it increased by 489%. Something similar is happening in other Latin American countries (Figure 1a, b), where the incidence and mortality of CRC are rapidly increasing due to a more sedentary lifestyle and bad eating habits. At the same time, the USA and Canada applied early screening detection strategies that appear to help lead to a steady state of deaths by CRC. Thus, the quality of endoscopic examinations as well as its early application should be improved in Latin American countries as a step forward to detect CRC at earlier stages and reduce its incidence and mortality.

Although in Mexico the proper screening for an early diagnosis and adequate and timely treatment of CRC is the responsibility of the state health care system, research support for the discovery of biomarkers and the development of novel therapies will help to improve this responsibility.

In this review, we have summarized the recent findings made by the Mex-CCRC, a multidisciplinary group of Mexican researchers focused on CRC research. Our first goal was to deepen our understanding of three aspects of the disease: the etiology, diagnosis and prognosis, and therapy at the basic biomedical level to broaden the knowledge of this disease.

## 2. Creation of the Consortium

Around 2013, basic and clinical research on CRC was almost nonexistent in Mexico. A literature search covering a thirty-year period (1983–2013) on CRC studies resulted in only a few papers originating in Mexico. Most case reports were from Mexican hospitals, and a few original research studies were found. Thus, a growing and important disease such as CRC has fueled our group to generate new knowledge in this field. Therefore, since 2013, a group of researchers have decided to work on some projects to shed light on basic biomedical research on CRC, thus consolidating the MEX-CCRC, which is working to help understand the mechanisms involved in the development and prevalence of CRC, from risk factors to possible new therapies (Appendix A). In this review, we discuss our contributions to date, and we address several factors associated with CRC development from different points of view. We are currently venturing into the study with human patients, where in some cases we want to apply and/or deepen the findings of our research.

## 3. Background

The factors that give rise to CRC are multiple and can be extrinsic, such as lifestyle, and intrinsic, such as the irreversible accumulation of new mutations, preexisting germline mutations, or chronic and persistent inflammation [3].

However, more than half of CRC-related cases and deaths are attributable to modifiable risk factors, such as smoking, diets high in processed foods, alcohol consumption, physical inactivity, and being overweight [4]. In addition, chronic inflammatory conditions, characterized by excessive recruitment and activation of cells of the immune system, such as inflammatory bowel diseases (IBDs), which includes ulcerative colitis, increase the risk of developing CRC by 2–3-fold [5,6,7].

It is widely accepted that CRC occurs after healthy tissue progresses to precancerous polyps that can give rise to the malignant tumor characteristic of CRC. It is important to note that mutations in the APC gene initiate 80–90% of CRC polyps and tumors [8]. Nonetheless, the exact pathways involved in polyp-to-tumor transformation are not fully understood [9]. At the onset and early stages of CRC, it is asymptomatic; nevertheless, by the time symptoms manifest, most patients are at advanced stages of the disease, with a worse prognosis than at early stages.

Approximately 25% of patients are diagnosed with CRC at a locally advanced stage, and close to 50% will develop metastasis, resulting in difficulties in treatment and subsequent CRC-related deaths [10,11]. Importantly, throughout the world, most diagnostic programs are based on the detection of polyps or tumors in the colon [12]. Therefore, thee early detection and timely removal of precancerous polyps prevent metastasis, reduce mortality, and improve prognosis and quality of life [13].

Different treatments based on cytotoxic chemotherapies have been implemented to combat CRC by suppressing tumor growth and metastasis. Despite this, the likelihood of patient survival for more than five years is low due to the prevalence of innate or acquired tumor resistance, which occurs in approximately 90% of patients with metastatic cancer [14,15].

## 4. The Mexican Colorectal Cancer Research Consortium Contributions

### 4.1. Etiology

#### 4.1.1. Microbiota

Recent studies have focused on evaluating the effect of manipulating nutrients in the diet and how they might be involved in preventing and treating CRC [16].

For example, dietary preferences influence factors that alter the host microbiota composition [17]. Thus, diet management by including foods such as fruits, whole grains, dairy products, and vegetables may reduce CRC incidence worldwide because there is enough evidence of its preventive role in colitis-associated colon cancer (CAC) development [18]. The Mediterranean diet includes most of the foods mentioned above and has been associated with a decreased risk of developing CRC (Figure 2a) [19].

On one hand, the analysis of the intestinal microbiome of healthy donors consuming a Mediterranean diet revealed that their microbiota has an increased abundance of the phyla *Verrucomicrobia* and *Bacteroidetes,* which favor an anti-inflammatory microenvironment. On the other hand, lower quantities of the phyla *Firmicutes*, *Euryarchaeota*, and *Fusobacteria* were detected in the intestinal microbiome of these healthy donors. In contrast, patients with inflammatory bowel disease (IBD) or CRC displayed a higher abundance of microbiota members that promote a proinflammatory environment, including the phyla *Proteobacteria*, *Fusobacteria*, and *Euryarchaeota.* Moreover, patients with IBD exhibited a higher abundance of *Firmicutes* and *Actinobacteria* but a lower abundance of *Verrucomicrobia* than healthy subjects [20]. Furthermore, these microbiota composition patterns were similar in patients with common variable immunodeficiency, where higher amounts of proinflammatory phyla (*Firmicutes*, *Actinobacteria*, and *Verrucomicrobia*) were associated with intestinal dysbiosis [21]. These findings highlight the relevance of dietary habits in the relationship between the immune system and the intestinal microbiota.

#### 4.1.2. Obesity

One of the main risk factors for developing CRC in both men and women is dietary habits that, among other alterations, increase the incidence of overweight and obesity. Obesity is considered an epidemic in several developed countries, becoming a disease of concern since it is linked to an increased risk of suffering from cardiovascular disease, diabetes, and several types of cancer, among which CRC stands out. However, this association appears to be stronger among men [22].

Although a body mass index (BMI) greater than 25 (>25) is associated with a higher risk of developing CRC [23,24], it has been reported that the survival of patients with gastric cancers is independent of BMI. In fact, patients with BMI > 25 had better prognoses and survival rates than those with BMI < 25, suggesting the participation of more complex mechanisms than nutritional indexes associated with cancer mortality [25]. Recent studies indicate that overweight and obesity could lead to CRC due to a chronic state of inflammation favored by dysregulated adipokines, inflammatory mediators, and other factors, such as immune cell infiltration [26]. However, the host’s and microbiota’s relationship are not entirely understood.

#### 4.1.3. Food Additives

##### Food-Grade Titanium Dioxide Enhances Tumor Formation in the Colon

Food-grade titanium dioxide (TiO2) is a mixture of micro- and nanoparticles used as one of the most common food additives in the food industry as a colorant for soups, cheeses, sauces, and bakeries, as well as pharmaceuticals [27]. However, this food additive has been used as an industrial pigment, mainly in paints, for over six decades. Recently, nanotechnology has decreased the titanium dioxide particle size, leading to the usage of nanosized TiO2. Unfortunately, nanosized particles can reach deeper areas of the respiratory tract, leading to bloodstream translocation. Sustained inflammation and oxidative stress are the main contributors to tissue alterations in experimental models of inhalation. In 2009, the International Agency for Research in Cancer (IARC) classified TiO2 (also called E171) in the 2B group as a possible carcinogen to humans by inhalation. Then, concern about the adverse effects of food-grade TiO2 arrived immediately in the research community. Our Consortium investigated whether E171 oral exposure might exacerbate colon tumor formation using the CAC murine model. In this model, we mimic the conditions in which a human might have a preexistent intestinal disease caused by genetic factors or a lifestyle in which high-fat or high-carbohydrate diets might promote proinflammatory environments associated with colon tumor development (Figure 2a).

Using the AOM/DSS CAC model, we demonstrated that E171 administered at a human equivalent dose (5 mg/kg/day) increased the expression of tumor progression markers in the colon of mice with CRC [28]. Such a report was the first evidence of E171 as an enhancer of tumors in the colon. Later, Bettini and his group demonstrated that the administration of E171 for 100 days induced colonic preneoplastic lesions in the AOM/DSS model [29]. More recently, we evaluated gene deregulation, and we discovered that E171 induced alterations in the expression of genes involved in the immune system response, oxidative stress, and DNA repair, as well as genes involved in the development of cancers, such as CRC [30]. Moreover, one of the key findings was the chromosomal damage evidenced by the E171-induced micronuclei in colon cells [31].

Based on the evidence mentioned earlier and in additional studies, the French Agency for Food and the Environment and Occupational Health Safety (ANSES) banned this food additive in 2019. Furthermore, the Netherlands Food and Consumer Product Safety Authority (NVWA) delivered an opinion on the possible health effects on the immune system of excessive consumption of TiO2. In 2020, the European Food Safety Authority (EFSA) indicated that E171 might no longer be considered safe when used as a food additive due to its genotoxic potential.

Our group also investigated some other adverse effects of oral consumption of E171, and we observed anxiety-like behavior. We found E171 to be an enhancer of liver disease induced by a high-fat diet [32]. Thus, the mechanism by which E171 enhances pathologies such as cancer needs to be completely elucidated.

### 4.2. Diagnosis and Prognosis

#### 4.2.1. Early Detection of CRC by Liquid Biopsy

Undoubtedly, one of the critical reasons for the increasing rise of CRC worldwide is the difficulty in detecting it early and the painful and expensive procedures available until now, such as colonoscopy, histopathological analysis, and imaging studies. Nevertheless, these techniques still do not fully satisfy clinical needs due to their low sensitivity and specificity in detecting early colorectal tumors.

Our group recently proposed using liquid biopsies to search for circulating free DNA (cfDNA) in blood plasma. This approach might allow early detection and a comprehensive molecular profile of CRC with minimum invasion on the patient. Taking advantage of the murine CAC model, our team performed cfDNA analysis to detect pathogenic alleles and mutations associated with CRC. Mutations at Ctnnb1 and Kras were detected using cfDNA at the early stages of colon tumor development, corresponding to the formation of aberrant crypt foci, the earliest histological alterations identified in CRC. These alterations using cfDNA were detected even earlier than microPET/CT imaging tests. Thus, it is possible to detect somatic mutations related to CRC at very early times by liquid biopsy; this technique may improve the diagnosis of CRC [33].

#### 4.2.2. A Role for the Immune Response during CRC

As the immune response was proposed as an emerging hallmark of cancer [34], emphasis has been placed on its role during some oncological pathologies, when the immunotherapy developed over some negative regulators of the immune activation (immune checkpoints) expressed in the tumor microenvironment has a leading role [35].

As a Consortium, we are interested in investigating the role of the immune response in CRC; for this purpose, we used the CAC murine model.

#### 4.2.3. Macrophage Migration Inhibitory Factor (MIF) Is a Crucial Regulator of CRC Initiation

Numerous studies have focused on comprehending how inflamed tissue becomes malignant. Among other hypotheses, it has been suggested that the nature of infiltrating immune cells and the soluble factors they secrete, such as cytokines and chemokines, are critical factors that can modify the cells adjacent to the inflammatory microenvironment [36].

In chronic inflammatory processes, such as IBDs, the proinflammatory cytokine macrophage migration inhibitory factor (MIF) is significantly overexpressed, and its levels are restored throughout the development of this pathology [37]. This increased MIF production has led to the proposal that this cytokine plays a crucial role in promoting CAC progression [38]. There is evidence that MIF acts as a chemokine that facilitates the recruitment of immune cells to the tumor site. For example, Mif knockout (*Mif^−/−^*) mice implanted with the CRC-derived cell line CT-26 developed smaller tumors than wild-type mice. Notably, although the mouse lacked MIF, the implanted CT-26 cells could produce large amounts of MIF. The presence of MIF was associated with increased recruitment of CD8+ T cells to the tumor site, thus establishing a regulatory role for MIF in T-cell trafficking [39]. These findings are in line with what is observed in patients with CRC: those with higher concentrations of MIF in connective tissue have survival rates longer than five years and significantly better survival than those patients with reduced MIF levels [40].

Our original work found that female *Mif^−/−^* mice with chemically induced CAC (no implantation of tumor cells) developed more colon tumors. However, we found lower frequencies of macrophages recruited to intestinal epithelial tumors than in wild-type mice. These results supported that MIF plays a role in controlling the initial development of CAC by attracting macrophages to the tumor, which is a condition that may favor the initial antitumor response (Figure 2c) [41].

Findings in which MIF appears to control CAC development and findings where MIF may promote malignancy and metastasis seem contradictory. Nonetheless, this discrepancy could be solved if these observations were confirmed and extended at different stages of CAC development, indicating whether the role of MIF differs according to the temporal progression of tumorigenesis. We suggest that MIF could suppress the development of CAC early in its development, while it is involved in malignancy and metastasis if CAC is already established. The role of MIF in promoting CAC has already been reviewed [42]. Therefore, we propose a dual role for MIF during the development of CAC, and currently, we are working on testing this hypothesis.

#### 4.2.4. A Role for STAT (Signal Transducer and Activator of Transcription) Signaling on CRC Initiation and Progression

Cytokines facilitate communication between immune cells, cancer cells, and nontransformed stromal cells inside the tumor microenvironment. A pivotal role for signal transducer and activator of transcription (STAT) proteins in initiating malignant transformation and tumor establishment has been reported in experimental models and cancer patients [43]. Tumoral levels of phosphorylated STAT6 (p-STAT6) have been frequently detected in the colon of patients with clinically detectable IBD, and tumoral p-STAT6 positively correlates with the clinical stage and poor prognosis of human CRC [44]. Activation of the STAT6 signaling pathway is critical for the recruitment of immune cells and plays a central role in tumor initiation and progression. Inhibition of STAT6 function in colitis or tumor cells may present a novel strategy for treating CRC [45].

In a murine model of colitis carcinogenesis, STAT6-deficient (Stat6−/−) mice exhibited fewer and less aggressive tumors, which was associated with a decrease in the mobilization of inflammatory monocytes (CD11b+Ly6Chi) and granulocytes (CD11b+Ly6G+) compared with wild-type animals. Moreover, the absence of STAT6 also impaired the expression levels of the inflammatory cytokines IL-17A and TNF-α [46].

Notably, we reported for the first time that STAT6 is critical for the induction of regulatory T cells (Tregs) (CD4+CD25+Foxp3+), which are capable of suppressing the antitumor response mediated by CD4+/CD8+ T cells and controlling the initial stages of CAC [47].

We demonstrated the importance of the balance of proinflammatory cytokines in controlling the development of CRC using knockout mice for STAT1 (STAT1−/−)—the signaling pathway for several proinflammatory cytokines—in the AOM/DSS model of CAC, by observing that it favored rapid and extensive intestinal damage and increased cell proliferation, leading to the early appearance of tumors [48]. These findings were linked to dysregulation of the recruitment of myeloid cells by IL 17, which led to increased recruitment of neutrophils and, consequently, an exacerbated inflammatory microenvironment (Figure 2b) [49]. Thus, the STAT1 and STAT6 signaling pathways may constitute viable targets for therapeutic intervention during the initial stages of CRC.

Our studies have emphasized that T-cell-associated responses are critical in CRC biology. Although the tumoral microenvironment influences T-cell functions, there is growing evidence that resident microorganisms modulate host immunity in CRC. For instance, the gut microbiota composition influences the frequencies of lamina propria cells such as Tregs [50] and T helper (Th) 17 cells [51]. Changes in the frequencies of Tregs and Th17 cells can promote carcinogenesis [52]. In line with these findings, it has been shown that the expression of the class I T-cell-associated molecule (CRTAM) on intraepithelial and lamina propria T cells is essential for maintaining intestinal microbiota homeostasis, as well as the Th17 immune response during inflammation (Figure 3a) [53]. Moreover, triple blockade of EGFR (epidermal growth factor receptor), MEK (mitogen-activated protein kinase kinase) and PD-L1 induces overexpression of CRTAM and has antitumor activity [54]. The precise mechanisms by which the interaction between the immune system and the host microbiota contributes to CRC development are not fully understood; therefore, studying the cellular and molecular mechanisms that underlie the development of this pathology is of utmost importance.

#### 4.2.5. Immunosuppression in the CAC Model

Previous studies suggested that Treg cells induce a tumor-promoting environment during CRC, as clinical samples from either blood or tumors showed increased Treg levels. Some experiments developed in both CAC and adenomatous polyposis coli (APC) Min mice suggested that ablation of Treg cells can induce an antitumor response; however, an exacerbated hyperinflammatory immune response developed in both cases [35,55]. These reports suggest that both the time and type of target for Treg cell depletion are essential to induce protection in a CAC murine model. We first detected that Treg cells have an increased suppressive profile during CAC by overexpressing Tim3, PD1, CD127, and CD25. We also found that Treg cells isolated from chronic CAC mice showed increased suppressive capacity over T-CD4+ and T-CD8+ cells. Finally, during the initial steps of CAC model development, we used an anti-CD25 monoclonal antibody for Treg cell depletion, proving that this methodology mainly reduced Treg cells but not activated T cells. When Treg cells were temporarily depleted there was a reduction in colon tumor development (Figure 3c) [56]. In addition, we also proposed that during CRC, there are subpopulations of regulatory T cells that could generate a favorable prognosis toward the development of CRC, depending on the co-expression of different surface, intracellular and intranuclear markers, in either the tumor or in the bloodstream [57]. It is necessary to analyze the role of Treg cells during the full development of CRC to better define its importance in this type of cancer.

In another approach, using genetically modified mice lacking one of the master receptors for the polarization of the anti-inflammatory immune response, such as the interleukin 4 receptor α chain (IL4Rα), we observed an increased and sustained proinflammatory response, resulting in a reduced colonic tumor burden [58]. To demonstrate whether macrophages have a role in this protection, we exclusively depleted the IL-4Rα receptor in macrophages, which display only an M1 classic profile. We observed that total polarization to the M1 profile in macrophages induces increased colon tumorigenesis (Figure 2c) [58]. These results strongly suggest that a balanced immune response is necessary to induce protection, with the coexistence of proinflammatory and anti-inflammatory responses.

### 4.3. Novel Therapies Development

#### 4.3.1. Chemical Therapies

Although CRC is curable when diagnosed early, most patients receive their diagnoses in advanced stages. Hence, treatments in advanced stages involve chemotherapy with severe adverse effects. Given the severity of chemotherapy, there is a growing interest in developing new compounds to treat CRC that may be less aggressive with patients [59,60]. In recent years, several studies have focused on metabolism and autophagy in cancer to develop efficient and specific treatments that limit severe side effects and, consequently, improve patient prognosis [61].

#### 4.3.2. Traditional Mexican Medicine: *Cyrtocarpa procera*

As mentioned above, inflammation is part of the immune response, which might damage healthy tissue, but when it is adequately regulated, it is beneficial for the host. Nevertheless, chronic inflammation is now recognized as significantly increasing cancer risk. Hence, strategies focused on reducing chronic inflammation could prevent or delay the onset of cancer [62]. For example, in a murine model of chronic colitis, the administration of the methanolic extract of *Cyrtocarpa procera (C. procera)*, a plant used in traditional Mexican medicine, decreased the clinical symptoms associated with colitis as well as the levels of chronic inflammation [63]. In addition to *C. procera*, several traditionally used plants have shown cytotoxic activity in human CRC cell lines in vitro (Figure 3b) [61]. In another study, in vivo treatment with the cellular extract of the yeast *Debaryomyces hansenii* effectively reduced the oxidative damage caused by chemically induced colitis in murine liver samples (Figure 3b) [64]. These findings pave the way for the continued exploration of new therapies based on natural products and the identification and isolation of the most effective molecules.

#### 4.3.3. Drug Repositioning

Recently, the number of new drugs undergoing preclinical testing has dropped, encouraging drug repurposing [65]. For instance, metformin, the most common drug used to treat type 2 diabetes, was experimentally used in combination with standard anticancer drugs, such as doxorubicin and sodium oxamate—triple therapy, Tt—to treat CAC. This triple therapy improved each drug’s individual or single benefit since, when combined, it significantly inhibited colon tumor growth and pro and anti-inflammatory cytokines, even in advanced stages of colitis-associated tumorigenesis (Figure 4a) [66]. A different approach from our group using the natural and nontoxic substance trimethylglycine, also known as betaine, was evaluated as an anti-inflammatory and nonconventional adjuvant with an inhibitor of STAT6 phosphorylation (AS1517499). This combined therapy improved the effect of 5-FU in advanced stages of CAC, reducing the number of tumors in the colon through apoptosis-induction (Figure 4b) and decreasing the levels of proteins related to malignancy in tumor cells at late stages, such as IL-10, TGF-β, IL-17A, SNAIL, and reducing β-catenin translocation to the nucleus. This combined therapy promotes a more significant response to 5-FU than individual therapies [67]. Together, these results show that inhibiting inflammation in a controlled way may help fight colitis-associated tumorigenesis.

#### 4.3.4. Targeting microRNAs (miRNAs)

MicroRNAs (miRNAs) are short single-stranded RNAs between 18 and 25 nucleotides in length that participate in biological processes through the negative regulation of mRNA transcripts, which have essential roles in different functions, such as cancer [68]. Due to their relevant participation in this pathology, there is growing interest in understanding the impact of miRNAs on cancer development. For example, in vitro administration of Tt induces apoptosis and autophagy by downregulating miR-26a in the human colorectal cancer cell line HCT116 (Figure 4a) [69]; in addition, miR-26a acts as a downregulator of the tumor suppressor gene Rb1 and promotes cell proliferation, metabolism, and migration in CRC [70]. Therefore, miR-26a may be a biomarker for this disease [71]. On the other hand, this combination of drugs decreases the expression of miR-106a in CRC cells, resulting in the induction of autophagy and subsequent apoptosis [72]. Even infection with *Entamoeba histolytica* triggers apoptosis of colon epithelial cells by inducing miR-643, which inhibits the antiapoptotic protein XIAP [73]. In line with this, SOX9 silencing potentiates the proapoptotic effect generated by BCL21 inhibition in poorly differentiated CRC cell lines [14]. This finding demonstrates a role for miRNAs in controlling tumor development and apoptosis in colon cells.

Proinflammatory cytokines activate cell proliferation and migration pathways, establishing a link between inflammation and CRC progression [74]. Several lines of evidence demonstrate that miRNAs regulate pathways involved in cytokine signaling; in this way, miRNA deregulation has a significant role in the development of pathologies such as IBD and CRC [75]. In line with this, recently, using a database, we found some miRNAs differentially expressed in healthy and inflamed colon tissue, where miR-30b-5p and miR-3065-5p correlated with decreased overall survival. In addition, in a CAC model, we found differential expression of miR21-5p, miR30b-5p, miR215-5p, mir3,065-5p and miR155-5p (Figure 2d) in tumors compared to inflamed or healthy colon tissue. These findings suggest miR-30b-5p and miR-3065-5p as prognostic biomarkers of CRC.

#### 4.3.5. Metabolic Blockade Therapy: Autophagy

Autophagy is an intracellular process that delivers cytoplasmic macromolecules, aggregated proteins, damaged organelles, or pathogens for digestion in lysosomes, generating single nucleotides, amino acids, sugars, fatty acids, and ATP for reuse in the cytosol [76]. This degradative process can sustain cell metabolism and survival during stress and preserve protein and organelle quality and quantity while eliminating damaged organelles and proteins [77]. Therefore, autophagy can prevent cancer and inflammation. In CAC, autophagy reduces excessive ROS produced during inflammatory responses, increases antibacterial defense, and decreases permeability of intestinal epithelial tissue, thereby inhibiting colitis and CRC onset [78]. However, although autophagy can favor tumor suppression by reducing damaged cellular components, it can also promote its progression by regulating the homeostasis of cancer stem cells [77]. In addition, it has been reported that cancer stem cells contribute to cancer development by increasing the blood supply to the tumor or favoring resistance to treatments; previously, these mechanisms have been extensively reviewed for gastrointestinal cancers [79]. Remarkably, in CRC, it has been proposed that autophagy is one of the critical factors that promote drug resistance, limiting treatment efficacy [80]. Thus, the above findings have prompted the quest for new therapies or repositioning existing therapies that target autophagy. In recent years, several drugs, such as aspirin, chloroquine, its derivatives, and niclosamide, have demonstrated anticancer properties, inhibiting the β-catenin pathway, autophagy, or both. These treatments and others that target these mechanisms are detailed elsewhere [81].

#### 4.3.6. Excreted-Secreted Products of Helminths

In addition to risk factors associated with lifestyle habits, an estimated 25% of cancers are related to chronic inflammation. Thus, a strong association between chronic inflammation and cancer has mainly been recognized [82]. For instance, in IBD, the development of CRC is one of the leading causes of mortality [83], highlighting the importance of understanding the involvement of inflammation in the tumor microenvironment, which has led to proposing anti-inflammatory therapies associated with long-lasting antitumor responses [84].

The inflammatory process may be a mechanism that allows some damaged protumoral cells to be eliminated. However, as seen in IBD, chronic inflammation promotes colon tumorigenesis [45]. In cases of chronic inflammation, it has been proposed that anti-inflammatory treatments could reduce the risk of developing some types of cancer. Consequently, new biological therapies focus on modulating inflammatory mediators [85].

Recent studies have examined helminths’ ability to modulate exacerbated inflammatory responses [86]. However, the role of helminth infections or their products in developing colorectal cancer is controversial. Currently, there is no certainty on how helminth immunomodulation impacts CRC incidence and progression. Several helminth-derived compounds have anti-inflammatory potential, although some intestinal helminth infections can worsen CAC development [87]. At least two reports indicated that intestinal colonization of *Heligmosomoides polygyrus* [88] or chronic *Trichuris muris* infection [89] could favor experimental CAC. In contrast, live preinfection with the helminth *Taenia crassiceps* decreased the number of colon tumors and consequently delayed cancer development in a similar mouse model of CAC [90].

Moreover, recent evidence from in vitro experiments using antigens derived from *H. polygyrus* significantly decreased murine and human CRC-derived cell lines proliferation, which was associated with increased expression of p53 and p21 [91]. In line with this, our research group was the pioneer in demonstrating that excreted-secreted products of *T. crassiceps* (TcES), instead of a whole infection, reduced in vivo colon tumorigenesis as a therapy delivered at the early stages of CAC (Figure 4b). This effect seems to be mediated by suppressing the STAT3 and NF-κB signaling pathways, which trigger proinflammatory and protumor signals [92]. Some of these beneficial features of helminths have already been discussed in detail [93]. These observations provide evidence that novel therapeutic strategies to combat CRC are necessary.

## 5. Concluding Remarks

Our findings reveal that microbiota disturbances, obesity, oral consumption of food additives, and deregulation of STAT1/6 signaling might be etiological components of CRC development. We also found that the MIF cytokine, miR26a, a panel of miRNAs associated with inflammatory regulation, and liquid biopsies detecting early mutations could be potential biomarkers of CRC diagnosis and prognosis. Among innovative therapies, we have suggested *C. procera*, a plant used in traditional Mexican medicine, the inhibition of STAT6 in combination with a standard chemotherapeutic agent, the use of trimethylglycine and the excreted-secreted products of helminths as possible alternative adjuvant therapies, given their ability to downregulate inflammation. Altogether, it has contributed to expanding the knowledge about the genesis and development of this critical pathology. Furthermore, this intense research has also contributed to the training and formation of young scientists to obtain different degrees such as bachelor’s, master’s, and Ph.D., contributing in this way to the development of scientific staff in Mexico.

## 6. Future Directions

The basic knowledge we have generated as a Mex-CCRC has led us to propose two more objectives in our project: first to translate some of the findings in the preclinical trials into pilot studies in Mexican patients with CRC (Figure 5), and second, to look for opportunities to interact with other organizations interested in CRC prevention and cure.

In addition, the incidence of CRC in our country is increasing, as in the rest of the world, but with essential differences in (a) sex, (b) geography, and (c) access to health insurance. We identify this as an opportunity to fuel or increase new research targets, diagnostic and prognostic strategies, and innovative therapies.

## Figures and Tables

**Figure 1 ijms-24-02115-f001:**
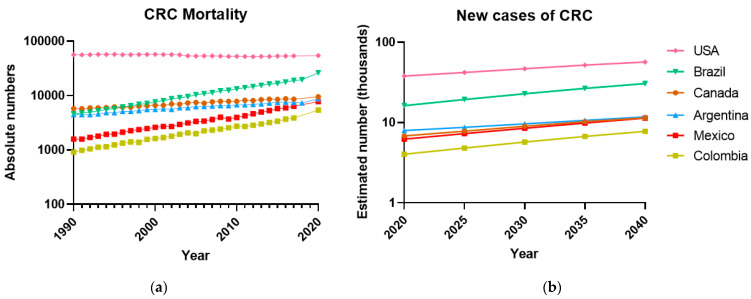
The Colorectal Cancer has increased in the last 30 years in Latin American countries. (**a**) Deaths per year associated with CRC, both sexes, in some representative countries of the American continent, from 1990 to 2020, according to the last GLOBOCAN report [1]. Rates are shown on a log10 scale. (**b**) The number of new cases of Colorectal Cancer will increase over time in several countries of the American continent, projections estimated according to GLOBOCAN [2]. Data not available for Mexico, GLOBOCAN estimates the data from national mortality using mortality rates models; incidence derived from cancer registry data in neighboring countries.

**Figure 2 ijms-24-02115-f002:**
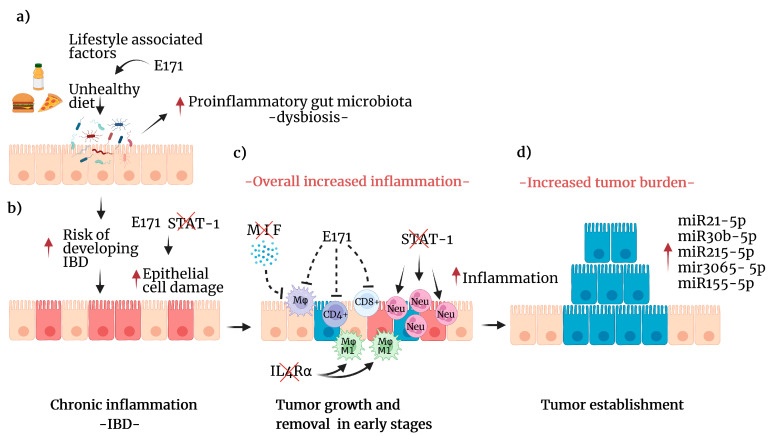
Lifestyle factors that increase the risk of developing CRC. (**a**) Effects of unhealthy diets include direct damage to epithelial cells, changes in the intestinal microbiota composition, and increases (red arrow) in proinflammatory bacteria leading to dysbiosis. These effects can be enhanced by food-grade titanium dioxide (E171). (**b**) Dysbiosis increases the potential of an IBD (red arrow), such as chronic colitis, in which chronic inflammation damages epithelial cells (red arrow). Additionally, E171 can affect the expression of genes associated with oxidative stress and DNA repair pathways. Furthermore, early STAT1 deficiency enhanced the damage and proliferation of colon cells. (**c**) The combined effect of this process attracts immune cells to detect and eliminate transformed cells (red). However, MIF deficiency reduces the number of macrophages recruited to the site, and E171 can impair the immune response. STAT1 deficiency leads to increased neutrophil infiltration, which exacerbates inflammation. Additionally, IL4Rα deficiency leads to M1 macrophage polarization, enhancing inflammation (red arrow). (**d**) Cancer initiation and promotion mechanisms may include the reduced detection of transformed cells by the concert of immune alterations (in blue), leading to enhanced cell proliferation and tumor growth. Furthermore, in this stage, several miRNAs are differentially expressed (red arrow) and linked to tumorigenesis. Created with BioRender.com, accessed on 27 June 2022.

**Figure 3 ijms-24-02115-f003:**
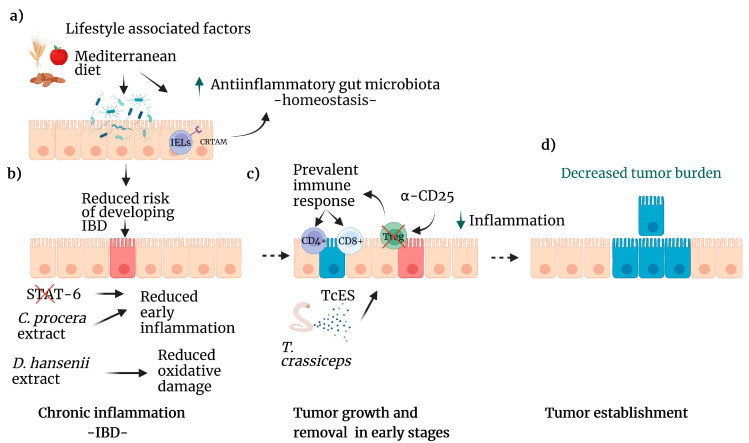
Lifestyle factors that reduce the risk of CRC. (**a**) Diets that include fruits, seeds, and whole grains as the Mediterranean diet improve the gut microbiota, increasing the population of anti-inflammatory bacteria (green arrow). CRTAM, an IELS receptor, is vital in maintaining homeostasis in the colon. This healthy microbiome reduces the risk of IBD. (**b**) Several natural compounds, such as *C. procera* extracts or STAT6 inhibitors, can reduce inflammation caused by IBD. *D. hansenii* extract diminishes oxidative damage caused by colitis. (**c**) Treg inhibition enhances the immune system response by T cell lymphocytes, and *T. crassiceps* excreted-secreted products modulate several pathways leading to decreased inflammation (green arrow). (**d**) As a result, tumor development decreases. Created with BioRender.com, accessed on 27 June 2022.

**Figure 4 ijms-24-02115-f004:**
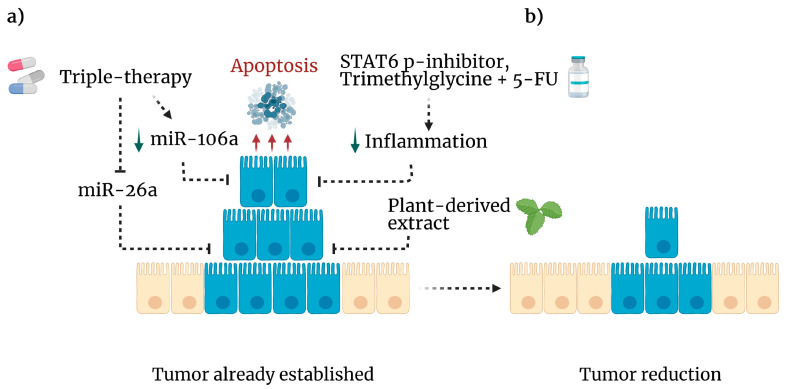
Experimental anticancer therapies. (**a**) If immune surveillance is unsuccessful, cancer cells proliferate to develop a tumor. However, several approaches have been tested to eliminate malignancy. Triple therapy (metformin, sodium oxamate and, doxorubicin) inhibits mIR-26a (dotted line) leading to apoptosis (red arrows) and downregulation of mIR-106a (green arrow). In addition, plant-derived compounds exhibit cytotoxic activity against CRC cell lines. A STAT-6 phosphorylation inhibitor plus trimethylglycine as adjuvant for 5-FU, diminishes inflammation (green arrow) and tumor cell viability. (**b**) As a result, these mechanisms reduced the tumor burden in mice. Created with BioRender.com, accessed on 27 June 2022.

**Figure 5 ijms-24-02115-f005:**
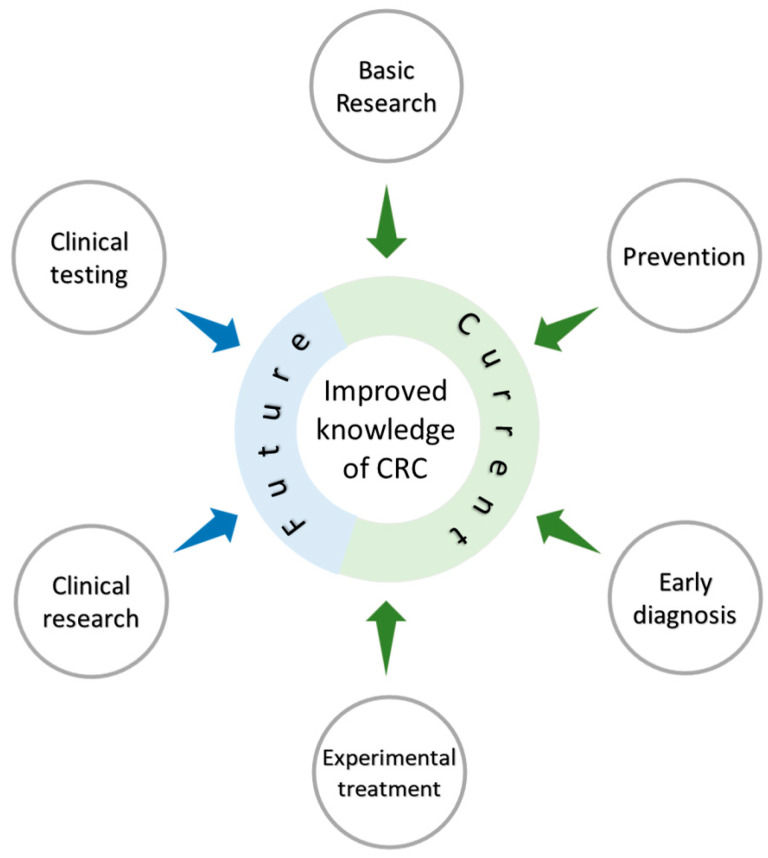
The Mexican Colorectal Cancer Research Consortium (MEX-CCRC) was developed to boost research on CRC in Mexico. The MEX-CCRC started impacting on the basic biomedical research of this field by characterizing immunological and molecular aspects of this pathology, and discovered biomarkers based on phenotypic characterization, which could ultimately lead to prevention and early diagnosis. Experimental treatments have also been tested with the possibility of doing in future clinical research that may result in the reduction and improvement of the disease.

## Data Availability

Data sharing not applicable. No new data were created or analyzed in this study.

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
