# Peer review of "Mexican Colorectal Cancer Research Consortium (MEX-CCRC): Etiology, Diagnosis/Prognosis, and Innovative Therapies"

_ijms, 2023, doi:10.3390/ijms24032115_

Round 1
Reviewer 1 Report (New Reviewer)
In the present review article, Andrade-Meza et al. summarized the etiology, diagnosis/prognosis and therapies for patients with colorectal cancer in several Latin American countries by comparing with North American countries. In the regard of colorectal carcinogenesis, they discussed several aspects including microbiota, food, cytokine signaling, immune cells, and the genetic alterations of miRNAs. Moreover, they prepared some good illustrations to explain their hypothesis for readers. Overall, this reviewer has no critical concerns. The following minor comments should be discussed in the revised version.
1. The quality of endoscopic examinations should be improved in Latin American countries. Thus, CRC should be found at earlier stage. Otherwise, the mortality of CRC is rapidly increased in those countries. Please explain this finding.
Author Response
In the present review article, Andrade-Meza et al. summarized the etiology, diagnosis/prognosis and therapies for patients with colorectal cancer in several Latin American countries by comparing with North American countries. In the regard of colorectal carcinogenesis, they discussed several aspects including microbiota, food, cytokine signaling, immune cells, and the genetic alterations of miRNAs. Moreover, they prepared some good illustrations to explain their hypothesis for readers. Overall, this reviewer has no critical concerns. The following minor comments should be discussed in the revised version.
- The quality of endoscopic examinations should be improved in Latin American countries. Thus, CRC should be found at earlier stage. Otherwise, the mortality of CRC is rapidly increased in those countries. Please explain this finding.
Response:
Thank you for the comments. We have attached a revised version of our manuscript. The comments were appropriate and enabled us to greatly improve the quality of our manuscript. We have highlighted the changes we made in blue font.
Your comment is very important, and we have included in the introduction section. Mainly because we agree that a better endoscopic technic, and also its diffusion and earlier application is critical in order to reduce the incidence and mortality of this disease. We have added this information.
Reviewer 2 Report (New Reviewer)

Author Response
1.Originality-the review focused on colorectal cancer research in Mexic analyzed is of continuous actuality, the Colorectal cancer is the second leading cause of death by cancer worldwide. 2. The objectives of the study are adequately described 4. The discussions are deepening of the studied problem. I think this paper is excellent and is an important addition to the literature. The review highlights important data related to epidemiology, pathophysiology, clinical appearance, and management of colorectal cancer in Mexic, focusing on clinical variants in the group, providing up-to-date information that has been overlooked in other articles.
Response
Thank you for the comments. We have attached a revised version of our manuscript. The English language and style were checked.
Reviewer 3 Report (New Reviewer)
The review by Andrade-Meza et al exposes an important initiative, the Mexican colorectal Cancer research Consortium (MEX-CCRC). As described, the Consortium represents the first organization to consolidate and promote basic and clinical research in the area of colorectal cancer in Mexico. The review is well written and generally easy to follow. A few very minor points should be considered that may further elevate this already important work. Independent of these suggestions, I recommend acceptance of this review and congratulate the authors on taking the inititive to create the MEX-CCRC.
1) In the abstract, the authors state that the 'Consoltium could be maintained as a national strategy'. As stated, it is not clear if the authors wish to say that this effort will continue in Mexico or if the continuation should be without the participation of other counties. The phrase then concludes 'the reported data can be translated inte other countries'. This is also unclear. Is the intention for the Consortium to serve as a model for other countries, but without opportunities for cooperation with other nations?
2) While point 1 above may be just a misunderstanding, I did not see any mention of opportunities and willingness for MEX-CCRC to interact with other organizations.
3) A few suggestions for the text:
line 65: 'conducted over thirty years ago' may be better as 'covering a thirty year period'
line 70: 'until conforming to the Mex-CCRC' may be better as 'thus consolidating the Mex-CCRC'
line 176: 'and some more studies' may be better as 'and in additional studies'
line 426: remove comma from 'sodium, oxamate'
Author Response
The review by Andrade-Meza et al exposes an important initiative, the Mexican colorectal Cancer research Consortium (MEX-CCRC). As described, the Consortium represents the first organization to consolidate and promote basic and clinical research in the area of colorectal cancer in Mexico. The review is well written and generally easy to follow. A few very minor points should be considered that may further elevate this already important work. Independent of these suggestions, I recommend acceptance of this review and congratulate the authors on taking the inititive to create the MEX-CCRC.
Response:
The comments were appropriate and enabled us to greatly improve the quality of our manuscript. We have highlighted the changes we made in blue font.
- In the abstract, the authors state that the 'Consoltium could be maintained as a national strategy'. As stated, it is not clear if the authors wish to say that this effort will continue in Mexico or if the continuation should be without the participation of other counties. The phrase then concludes 'the reported data can be translated inte other countries'. This is also unclear. Is the intention for the Consortium to serve as a model for other countries, but without opportunities for cooperation with other nations?
Response: We believe that the results of the various studies carried out by the MEX-CCRC consortium should be considered to influence the national health strategy in Mexico, and to establish future collaborations with other countries, considering the characteristics of the region. In particular, we believe that some of our observations could be extrapolated to some Latin American countries, which share similar habits and customs. We have added some lines related with your important comments in the abstract and future direction sections.
- While point 1 above may be just a misunderstanding, I did not see any mention of opportunities and willingness for MEX-CCRC to interact with other organizations.
Response: In this publication, in addition to publicizing the progress we have made as a consortium, it is opening the possibility of interacting with other scientific groups that are working in the area. We are in the best disposition and eager to interact with other organizations. We have added this intention in the future direction section.
3) A few suggestions for the text:
line 65: 'conducted over thirty years ago' may be better as 'covering a thirty year period'
line 70: 'until conforming to the Mex-CCRC' may be better as 'thus consolidating the Mex-CCRC'
line 176: 'and some more studies' may be better as 'and in additional studies'
line 426: remove comma from 'sodium, oxamate'
Thank you for the appropriate comments, all of them have been addressed.
This manuscript is a resubmission of an earlier submission. The following is a list of the peer review reports and author responses from that submission.
Round 1
Reviewer 1 Report
The authors focused on colorectal cancer, covering the majority of related topics whit a special interest in research results. Both the epidemiological and experimental data are widely treated and supported by a huge amount of literature. We observe a lack of information about an important topic i.e the role of polyps in CRC, both as regards the strategy prevention and the genetic implication in the neoplastic pathway. Of note, the majority of screening programs worldwide are based on polyp detection. Additionally, we suggested better defining the role of helminths in CRC pathogenesis.
Author Response
Thank you for the comments. We have attached a revised version of our manuscript. The comments were appropriate and enabled us to improve the quality of our manuscript significantly. We have highlighted the changes we made in blue font.
Q1. We observe a lack of information about an important topic i.e the role of polyps in CRC, both as regards the strategy prevention and the genetic implication in the neoplastic pathway. Of note, most screening programs worldwide are based on polyp detection.
Answer: In the introduction, we have added information about the role of polyps in CRC, lines 49 to 53; 58 to 59; 72 to 74.
It is widely accepted that CRC occurs after healthy tissue progresses to precancerous polyps that can give rise to the malignant tumor characteristic of CRC. It is important to note that mutations in the APC gene initiate 80-90% of CRC polyps and tumors (1). However, the exact pathways involved in polyp-to-tumor transformation are not fully understood (2). It is important to note that, throughout the world, most diagnostic programs are based on the detection of polyps or tumors in the colon. It should be noted that timely removal of precancerous polyps reduces patient mortality (3).
Q2. We suggested better defining the role of helminths in CRC pathogenesis.
Answer: We have included information about the role of helminths in CRC pathogenesis lines 412 to 415; 417 to 419.
The role of helminth infections or their products in developing colorectal cancer is controversial. Currently, there is no certainty on how helminth immunomodulation impacts colorectal cancer incidence and progression. As cited in this manuscript, some findings point out a detrimental effect of helminth infection on CRC development. At least two reports indicated that intestinal colonization of Heligmosomoides polygyrus (4) or chronic Trichuris muris infection (5) could favor experimental CRC. However, our group has two strong papers indicating that extraintestinal infection with the helminth Taenia crassiceps or its products actively inhibits tumor cell growth and progression of CRC in a murine model.
Moreover, treatment with T. crassiceps excreted/secreted products in mice with already ongoing CRC, dramatically reduced the number of colon tumors. In addition, in vitro experiments using antigens derived from H. polygyrus significantly decreased murine and human CRC cell proliferation, which was associated with increased expression of p53 and p21 (6). Furthermore, exposure to these antigens reduced murine and human CRC cell mitochondrial activity. Similarly, T. crassiceps-derived products downregulated STAT3 activity, associated with increased cell proliferation in tumor cells. Thus, further work is required to fully understand how helminths and their products can alter the development of colorectal cancer.
In addition, we added a section regarding our findings in miRNAs as potential targets or biomarkers for CRC.
Also, we have added a Concluding remarks section and a Future directions section with the following goals that our Consortium wants to pursue.
- Becker WR, Nevins SA, Chen DC, Chiu R, Horning AM, Guha TK, et al. Single-cell analyses define a continuum of cell state and composition changes in the malignant transformation of polyps to colorectal cancer. Nat Genet. 2022;54(7):985-95.
- Mangifesta M, Mancabelli L, Milani C, Gaiani F, de'Angelis N, de'Angelis GL, et al. Mucosal microbiota of intestinal polyps reveals putative biomarkers of colorectal cancer. Sci Rep. 2018;8(1):13974.
- Ladabaum U, Dominitz JA, Kahi C, Schoen RE. Strategies for Colorectal Cancer Screening. Gastroenterology. 2020;158(2):418-32.
- Pastille E, Frede A, McSorley HJ, Grab J, Adamczyk A, Kollenda S, et al. Intestinal helminth infection drives carcinogenesis in colitis-associated colon cancer. PLoS Pathog. 2017;13(9):e1006649.
- Hayes KS, Cliffe LJ, Bancroft AJ, Forman SP, Thompson S, Booth C, et al. Chronic Trichuris muris infection causes neoplastic change in the intestine and exacerbates tumour formation in APC min/+ mice. PLoS Negl Trop Dis. 2017;11(6):e0005708.
- Jacobs BA, Prince S, Smith KA. Gastrointestinal Nematode-Derived Antigens Alter Colorectal Cancer Cell Proliferation and Migration through Regulation of Cell Cycle and Epithelial-Mesenchymal Transition Proteins. Int J Mol Sci. 2020;21(21).

Reviewer 2 Report
A great introduction to state of direction of Mexican consortium. If this is within the scope of the journal (which I leave to the editors), I suppose it is a good insight into what is being done in Mexico about CRC.
However, I lack the distinction of what is your interest from what the consortium really does or did so far which would be interesting. Is this just a plan based on this review or there is translational research going on as we speak.
Sex differences were introduced on two occasions, but fail to be elaborated later on, if already stressed.
Overall, I miss the clear aim of the paper.
Author Response
Thank you for your comments. They are very appropriate and have served to improve the writing. We have added information that clarifies the objectives of the formation of the Mexican Colorectal Cancer Research Consortium (MEX-CCRC). We have also clarified that it is a report of our progress and the direction our work will take to get closer to patients with CRC .
Q1. I lack the distinction of what is your interest from what the consortium really does or did so far which would be interesting. Is this just a plan based on this review or there is translational research going on as we speak.
Answer. The consortium's objective was to bring together a group of multidisciplinary researchers who would focus their expertise to contribute to the understanding of CRC, which is an emerging health problem in the world, particularly in Mexico. Lines 68 to 74.
As a first approach to the study of CRC, we used in vitro models (cell cultures) and a murine model of CRC, which allowed us to contribute to the basic knowledge of this pathology. In this review, we report our first findings.
We have also highlighted that we continue to work on CRC; now, we are venturing into the study with human patients where, in some cases, we want to apply and/or deepen the findings of our previous research.
We have highlighted this in the concluding remarks and future directions section. Lines 432 to 445; 447 to 470, respectively.
Q2. Sex differences were introduced on two occasions but fail to be elaborated later on, if already stressed.
Answer. The incidence and mortality of CRC are different between men and women. Undoubtedly, the hormonal profile, as well as the lifestyle, are determining factors in these differences observed between men and women. As a consortium, we want to delve into this area of study. Section: Future directions, Lines 454 to 460.
Q3. Overall, I miss the clear aim of the paper.
Answer. The aim of this review has been highlighted in the introduction section. Lines 70 to 74.
In addition, we added a section regarding our findings in miRNAs as potential targets or biomarkers for CRC.
Also, we have added a Concluding remarks section and a Future directions section with the following goals that our Consortium wants to pursue.
